# Quantification of Death Risk in Relation to Sex, Pre-Existing Cardiovascular Diseases and Risk Factors in COVID-19 Patients: Let’s Take Stock and See Where We Are

**DOI:** 10.3390/jcm9092685

**Published:** 2020-08-19

**Authors:** Amalia Ioanna Moula, Linda Renata Micali, Francesco Matteucci, Fabiana Lucà, Carmelo Massimiliano Rao, Orlando Parise, Gianmarco Parise, Michele Massimo Gulizia, Sandro Gelsomino

**Affiliations:** 1Department of Cardiothoracic Surgery, Cardiovascular Research Institute Maastricht University, Universiteitssingel 50, 6229 ER Maastricht, The Netherlands; amaliamoula1@gmail.com (A.I.M.); l.micali@maastrichtuniversity.nl (L.R.M.); francesco.matteucci@maastrichtuniversity.nl (F.M.); o.parise@icloud.com (O.P.); g.parise@maastrichtuniversity.nl (G.P.); 2Grande Ospedale Metropolitano, 89124 Reggio Calabria, Italy; fabiana.luca92@gmail.com (F.L.); massimo.rao@libero.it (C.M.R.); 3Cardiogy Complex Unit, Garibaldi Nesima Hospital, 95122 Catania, Italy; michele.gulizia60@gmail.com; 4Heart Care Foundation Onlus, 50121 Florence, Italy

**Keywords:** COVID-19, SARS-CoV-2, cardiovascular diseases, mortality, hypertension, diabetes mellitus

## Abstract

Patients with pre-existing cardiovascular disease (CVD) might be more susceptible to infection from severe acute respiratory syndrome coronavirus 2 (SARS-CoV-2) and have higher mortality rates. Nevertheless, the risk of mortality has not been previously quantified. The aim of this meta-analysis is to quantify the risk of mortality in coronavirus disease 2019 (COVID-19) patients. A meta-analysis was conducted analyzing the impact of (1) sex, (2) age, (3) CVD with coronary artery disease (CAD), (4) CAD alone, (5) CVD without CAD, (6) hypertension, (7) cerebrovascular diseases, and (8) diabetes on mortality. Relative risk was assessed for dichotomous variables, mean difference for continuous variables. Twenty-six studies were included, encompassing 8497 patients. Males had 16% higher risk of mortality than females (*p* < 0.05) and elderly patients had higher chance of dying than younger patients (*p* < 0.0001). Patients with overall CVD have a 1.96-fold higher mortality risk (*p* < 0.0001). CAD increases risk of mortality by 1.90-fold (*p* < 0.05). CVD-CAD were found to increase risk up to 2.03-fold (*p* < 0.05). Hypertension, cerebrovascular disease and diabetes increase the risk of death up to 1.73-fold, 1.76-fold and 1.59-fold, respectively (*p* < 0.0001, *p* < 0.0001, *p* < 0.05, respectively). Sex, age, presence of CAD and/or other types of CVD, hypertension, cerebrovascular diseases and diabetes mellitus increase mortality in patients with COVID-19.

## 1. Introduction

Coronavirus disease 2019 (COVID-19) is an infectious disease caused by severe acute respiratory syndrome coronavirus 2 (SARS-CoV-2). The virus, first identified in 2019 in China, has a positive sense single stranded RNA and seems to be of zoonotic origin. The virus is most likely airborne and highly contagious. It spreads via contaminated droplets that pass from one human to another while in close contact [1].

The recent global pandemic ignited by the COVID-19 has had a considerable impact on many healthcare systems around the world [2]. For this reason, the disease has received increasing attention by the scientific community. Previous literature suggests that patients with pre-existing cardiovascular disease (CVD) might be potentially more susceptible to infection from SARS-CoV-2 [3]. Nevertheless, the exact mechanisms by which COVID-19 affects the cardiovascular system and mortality are not yet well understood, despite accumulating evidence that such a connection exists [4,5].

However, to the best of our knowledge the risk of death in relation to sex, age and CVD has not been quantified in large cohorts of patients. Moreover, the association between COVID-19, CVD and patient mortality has not yet been fully elucidated, setting the need for additional confirmation of the association between these two ailments as well as the impact of shared risk factors on such a relation. Therefore, the present meta-analysis is aimed to quantify the risk of mortality in relation to sex, age and pre-existing CVD in COVID-19 patients, and attempt to identify the potential factors involved in such a causation.

## 2. Materials and Methods

### 2.1. Search Strategy

The literature search was conducted in accordance with the principles of the Preferred Reporting Items for Systematic Reviews and Meta-Analyses (PRISMA) [6] and the Cochrane handbook [7]. Two authors established the search strategy (AIM and LRM) and the decisions were approved by a third author (SG).

One investigator performed the literature search (AIM), which was limited to articles published from 1 December 2019 until 18 May 2020. An unrestricted literature search was performed using PubMed, adopting the following search terms: “severe acute respiratory syndrome coronavirus 2” [Supplementary Concept] OR “COVID-19” [Supplementary Concept] OR “spike glycoprotein, COVID-19 virus” [Supplementary Concept] OR “covid-19” OR “covid 19” OR “COVID19” OR “SARS-CoV-2” OR “novel coronavirus” AND epidemiology OR comorbidities OR heart OR cardiovascular OR myocardial OR “Cardiovascular Diseases” [Mesh] OR heart OR myocardium OR STEMI OR infarction OR arrhythmia OR hypertension.

### 2.2. Selection Process

The articles were selected based on the following inclusion criteria: (1) human studies; (2) full articles about COVID-19 disease containing separate data for patients that survived and patients that did not; (3) analyses of fatality cases; (4) studies including at least 10 patients; (5) articles published from December 2019 and (6) articles in English language.

The exclusion criteria used to reject articles were: (1) non-human studies; (2) case reports; (3) previous reviews and/or meta-analyses; (4) editorials; (5) comments; (6) studies without separate data on surviving and non-surviving patients; (7) studies in languages other than English.

### 2.3. Risk of Bias Assessment

Two reviewers (AM and LM) independently assessed the risk of bias for the included studies. The ROBINS-I tool (Risk of Bias in No-Randomized Studies of Interventions) was used for the assessment of bias at the individual study level [7]. Disagreements were resolved by discussion or by involving a third reviewer (SG). The domains assessed were (1) bias due to confounding; (2) bias in selection of participants into the study; (3) bias in classification of interventions; (4) bias due to deviations from intended interventions; (5) bias due to missing data; (6) bias in measurement of outcomes; (7) bias in selection of the reported result; and (8) overall bias assessment. The evaluation of the aforementioned domains was conducted with the aid of Cochrane handbook [7]. Furthermore, the generation of the plot for ROBINS-I was achieved the software robvis [8].

### 2.4. Endpoints

The primary endpoint of this meta-analysis was to identify comorbidities and pre-existing cardio-metabolic diseases that could predict mortality in patients with COVID-19. The risk factors and comorbidities taken into consideration were: (1) sex, (2) age, (3) CVD, (4) coronary artery disease (CAD), (5) hypertension, (6) cerebrovascular disease, and (7) diabetes mellitus.

To evaluate the impact of the several types of CVD and CAD separately, first we conducted an analysis of patients with CVD including CAD (“CVD with CAD”) vs. patients without CVD; then a separate analysis only of patients with CAD (“CAD”) vs. patients without CAD; and finally, an analysis of patients with CVD excluding CAD (“CVD without CAD”) vs. patients without CVD.

### 2.5. Statistical Analysis

The meta-analysis was conducted using v. 3.6.1 (R Foundation for Statistical Computing, Vienna, Austria). Relative risk (RR) and 95% confidence interval (CI) were used as index statistics for dichotomous variables. For continuous variables, mean difference and 95% CI were calculated. In both cases the random effects model was adopted because heterogeneity among studies was anticipated. Heterogeneity was assessed with the statistical inconsistency Higgin’s I^2^ test [7]. I^2^ values < 40% were considered having low heterogeneity, I2 values > 75% were considered having high heterogeneity [7]. Publication bias was evaluated using Egger’s test of the intercept. *p* values < 0.05 were considered statistically significant.

## 3. Results

### 3.1. Search Results and Characteristics of the Studies

The initial search retrieved 1719 articles. After screening for the inclusion and exclusion criteria, 78 articles that included patient demographic data for COVID-19 and mortality were found. After rejecting articles without separate data for patients that survived and non-survivors, a total of 22 articles were found to fulfil the criteria. Four additional articles were added from the references of the articles found through the search on PubMed (Figure 1). At the end of the selection process, 26 studies were included in the analysis [9,10,11,12,13,14,15,16,17,18,19,20,21,22,23,24,25,26,27,28,29,30,31,32,33,34]. Eight out of the twenty-six studies were analyses of fatality cases [11,12,15,18,21,26,33,34]. There were twenty papers from China [9,10,11,12,13,14,15,16,18,22,23,24,25,26,27,28,29,30,31,32], two papers from Italy [17,21], two papers from Korea [33,34], one from Iran [20] and one from the United States of America [19].

In total, the patient cohort included of 8497 patients, of whom 5121 (60.3%) were male and 3376 (39.7%) were female. Non-survivors among males accounted for 25.0% of the patients (1280 individuals out of 5121), whereas non-survivors percentage among females was 19.5% (659 individuals out of 3376). The studies that were included in the meta-analysis and the characteristics of the patients with COVID-19 are shown in Table 1. 

### 3.2. Assessment of Bias

Figure 2 shows the “risk of bias” graph. Low bias due to confounding was absent in all papers, as confounding was expected in all of them, although some of them [10,14,15,16,19,21,22,23,24,25,29,33,34] controlled for confounding through either multivariable and multivariate analysis, adjustment and stratification of patients. Twelve papers [9,10,11,13,14,24,25,27,28,29,31,32] had low bias due to selection of participants, as all patients eligible were included in the study and were enrolled in the same short period of time. No paper had low bias for classification of interventions as intervention status was not well defined. In addition, for the bias due to deviations from intended intervention, no study had low bias mainly due to the fact that in the majority this information was missing. Two papers [11,29] show low bias due to missing data because they had complete data. Fifteen papers [9,10,11,13,14,16,17,24,25,26,27,28,29,30,31] had low bias in measurement of the outcome. Eight studies [16,19,23,24,25,30,31] had low bias in the reported results. No study was overall lowly biased.

### 3.3. Prognostic Factors for Mortality

Incidence rate ration (IRR), heterogeneity test and Egger’s test results are summarized in Table 2. The analysis revealed that men have 16% higher risk of mortality than women, as shown in Figure 3A (RR: 1.16 [95% CI: 1.05, 1.27], *p* = 0.003; funnel plot in Appendix A). As shown in Figure 3B, we found that age is another predictor of mortality, as older patients had significantly higher chance of dying than younger patients (mean difference: −15.72 [95% CI: −18.62, 12.81], *p* < 0.0001; funnel plot in Appendix A).

Furthermore, the analysis revealed that patients with CVD with CAD have a 1.96-fold [95% CI: 1.51, 2.54] higher risk of mortality than patients without CVD, as presented in Figure 4A (*p* < 0.0001; funnel plot in Appendix A). In particular, the presence of CAD increased the risk of mortality by 1.90-fold [95% CI: 1.32, 2.74] when compared to other CVDs, as described in Figure 4B (*p* = 0.0005; funnel plot in Appendix A). However, by conducting an analysis on CVD excluding CAD we found that the presence of other CVDs is a strong predictor of death, since patients with CVD (excluding CAD) had 2.03-fold [95% CI: 1.41, 2.92] higher risk of mortality compared to patients without CVD (Figure 4C, *p* = 0.0002; funnel plot in Appendix A). We found that non-survivors were more likely affected by hypertension, having a 1.73-fold greater risk of mortality than patients without hypertension (Figure 4D, RR: 1.73 [95% CI: 1.37, 2.19], *p* < 0.0001; funnel plot in Appendix A). Patients with pre-existing cerebrovascular disease tend to die 1.76-fold more than patients without cerebrovascular disease, suggesting that cerebrovascular disease is a strong predictor of death (Figure 4E, RR: 1.76 [95% CI: 1.25, 2.50], *p* < 0.0001; funnel plot in Appendix A). Similarly, as pictured in Figure 4F, patients with diabetes had 1.59-fold higher chance of dying than patients without diabetes (RR: 1.59 [95% CI: 1.25, 2.02], *p* < 0.0001; funnel plot in Appendix A).

## 4. Discussion

### 4.1. Increased Mortality in Males and Elderlies

In our analysis, we quantified the risk of death in almost 8500 COVID-19 patients in relation to sex, age, pre-existing CVD and cardiovascular risk factors. To the best of our knowledge, this has not been previously done, especially in cohorts encompassing large numbers of patients. Interestingly, our updated report shows that men still are more prone to dying but the effective increase in risk in males compared to females is lower than previously reported (around 16%). An increased risk of mortality for male COVID-19 patients (2.4 times that of women) has been widely reported. This disproportionate death ratio in men was explained by a higher incidence of pre-existing disease, higher risk behaviors, occupational exposure, high levels of androgens in men, and behavioral/social differences that favor women [35]. Androgens increase the expression of the transmembrane protease serine 2 (TMPRSS2) [36]. TMPRSS2 is a critical protease that enables the entry of SARS-CoV-2 in angiotensin-converting enzyme 2 (ACE2) receptors, explaining why men tend to die more from COVID-19 [36,37]. A treatment against androgens that could theoretically interfere with the course of the disease is still debated [36]. However, it must be considered that although higher male-to-female death ratio was confirmed in all the countries with available data, the United States with the largest reported outbreak of COVID-19 in the world provided only partial sex-disaggregated data and this might have biased the overall estimation of sex-related risk distributions. This warrants a careful epidemiological analysis to assess whether there has really been a turn in sex-specific differences with a rising incidence of death in women [38].

In contrast, the association with age was confirmed, with older patients being more vulnerable to die from COVID-19 [39]. Primarily responsible for the increased age-related susceptibility are the ACE2 receptors and CD26; both overexpressed in senescent cells. Both ACE2 receptors and CD26 are targets for coronaviruses, and their overexpression in older patients might mediate the increased fatality rate in COVID-19 patients [40]. ACE2 is abundantly distributed in the lungs but also in the heart, kidneys, guts, and the pancreas [41]. ACE2 is pivotal for the entry mechanism of the SARS-CoV-2 as it is harnessed by the virus as an entry point, whereas CD26 interacts with the S1 domain of the virus affecting virulence [42,43,44]. Another mechanism contributing to the increased mortality in elderly patients is immunosenescence, in which naïve T and B cells are produced in lower quantities, and dendritic cells do not effectively differentiate after T cell interaction [39].

### 4.2. Increased Mortality in Patients with Pre-Existing CVD

The third finding of our meta-analysis is that the presence of cardiovascular diseases, is associated with a higher risk of mortality when compared to COVID-19 patients without pre-existing CVD. Our outcomes are in contrast with the results of a previous meta-analysis conducted on three studies, which found no correlation between the history of CVD and mortality but revealed an association between CVD and enhanced disease severity [5]. It is possible that such a discrepancy could be due to the difference in terms of number of studies included in the analysis. Previous literature already suggests that CVD might be involved in promoting death in COVID-19 patients [45]. Another recent meta-analysis conducted on six papers reported that among COVID-19 patients admitted to the intensive care unit, 17.1% had hypertension and 16.4% cardiac and/or cerebrovascular diseases [3]. Wu et al. [46] also reported that patients with CVD, hypertension and diabetes tend to die more often. The cause of such an association might be complex and multifactorial. Cardiopathic patients with ventricular hypertrophy, diastolic dysfunction and heart failure tend to develop acute pulmonary hypertension while being affected by COVID-19. This can result in pulmonary edema [41]. If SARS-CoV-2 causes sepsis, then acute respiratory distress syndrome (ARDS) can occur which per se aggravates the edema, and can become the cause of death in these patients [41,47]. Additionally, when infection by SARS-CoV-2 occurs, the virus is internalized and this triggers the activation of ADAM metallopeptidase domain 17 (ADAM17). ADAM17 causes cleavage of the ACE2 receptors making them unresponsive to the negative feedback exerted by the activation of the renin–angiotensin–aldosterone system. This is ultimately responsible for further production of cytokines, which aggravate the inflammation [48]. In the presence of pre-existing CVD, the cytokine storm can exacerbate underlining diseases by aggravating pre-existing heart failure, causing depression of myocardial activity, increasing the oxygen demand/supply ratio and endothelial dysfunction [22,48]. In addition, 17% patients with COVID-19 had pre-existing CAD and this raises the risk of death, especially when it is associated with potential hypercoagulability deriving from the febrile state [41]. Nonetheless, in our analysis, CVD with or without CAD showed very close RRs of death. In other words, although the presence of CAD alone raised the risk of death by 90%, the presence of coronary disease did not increase death RR of patients with other CVDs that increases the risk > 100%. Moreover, we have found that COVID-19 patients with hypertension had a 73% higher RR than those without high blood pressure. Unfortunately, due to the lack of specific information, it was not possible to compare subgroups to study the true incremental risk associated with hypertension in COVID-19 patients with CVD. However, it has been proposed that when patients with heart failure and hypertension receive ACE inhibitors and type-I receptor blockers (ARBs), these agents contribute to the upregulation of ACE2 receptors. This increases susceptibility to contracting COVID-19. The mechanisms mentioned help explain their vulnerability to mortality. For this reason, some authors have suggested the use of alternative antihypertensive medication during the pandemic, such as calcium channel blockers [49].

### 4.3. Increased Mortality in Patients with Diabetes

The fourth finding of our analysis is the increased fatality rate in patients with diabetes. According to the meta-analysis conducted by Li et al. [3], patients with diabetes represented 9.7% of the COVID-19 patients in Intensive Care Unit (ICU). Susceptibility of diabetic patients towards COVID-19 and their increased chance of dying derives from the overexpression of ACE2, impaired innate immunity and delayed Th1 cell-mediated responses. These factors predispose to cytokine storm, with adverse outcomes. Furthermore, while on one hand, insulin reduces ACE2 expression, on the other, hypoglycemic drugs and statins upregulate ACE2 [50]. In addition, diabetic patients might need additional administration of insulin or secretagogues, as the viral infection can stimulate cortisol release and thus increase of blood glucose levels. However, these drugs alter water and sodium reabsorption and increase the risk of developing pulmonary edema in cardiopathic patients, especially if sepsis causes renal dysfunction. Therefore, intravenous fluids administration should also receive attention by clinicians. In this situation, concomitant treatment with ACE inhibitors can aggravate the load on the respiratory system. For this reason, some authors suggest careful evaluation of the status of the lungs and interruption of ACE inhibitors and ARBs, if necessary before ARDS manifests worsening the prognosis of patients [41].

## 5. Study Limitations

This study has some limitations. First, the majority of the studies were retrospective, predisposing to the risk of bias. Second, some of the studies included were analyses of fatality cases. Third, heterogeneity between studies was high in all the endpoints analyzed, because of a great variety in baseline characteristics. Fourth, it is possible that the definition of CVD could be different in the different hospital settings/countries, with most of the papers that were included not including detailed description of the type of cardiovascular disease of the patient. Fifth, it would have been of great interest to study the interaction between CVD and single risk factors in predicting death. Unfortunately, within the papers data were not split into sub-groups to allow these analyses. Finally, in the reports there was reported age cut-off, therefore it was not possible to examine the RR increase with age. The only data attainable was the difference in age between survivors and not survivors.

## 6. Conclusions

Our results provide a quantification of mortality risk in COVID-19 patients with pre-existing cardiovascular comorbidities. Our results demonstrate that sex, age, presence of CAD and/or other types of CVD, hypertension, cerebrovascular diseases and diabetes mellitus increase mortality in patients with COVID-19. In particular, CAD and/or other types of CVD, hypertension, cerebrovascular diseases almost double the risk of mortality. Further research to identify the underlining mechanisms of such an association is warranted.

## Figures and Tables

**Figure 1 jcm-09-02685-f001:**
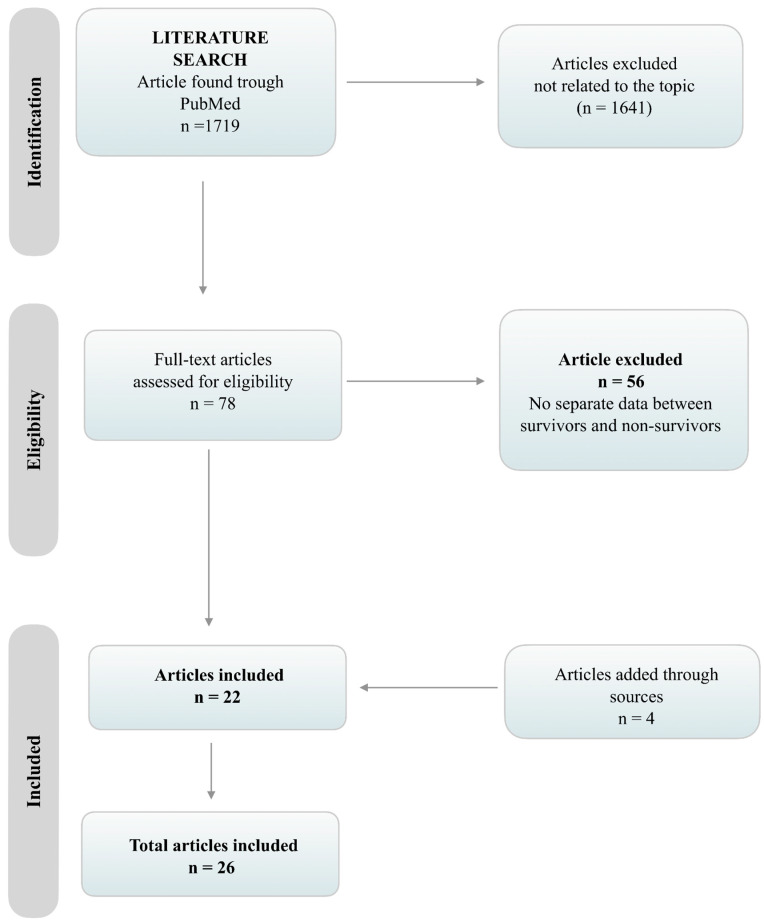
Preferred Reporting Items for Systematic Reviews and Meta-Analyses (PRISMA) flowchart of the selection process.

**Figure 2 jcm-09-02685-f002:**
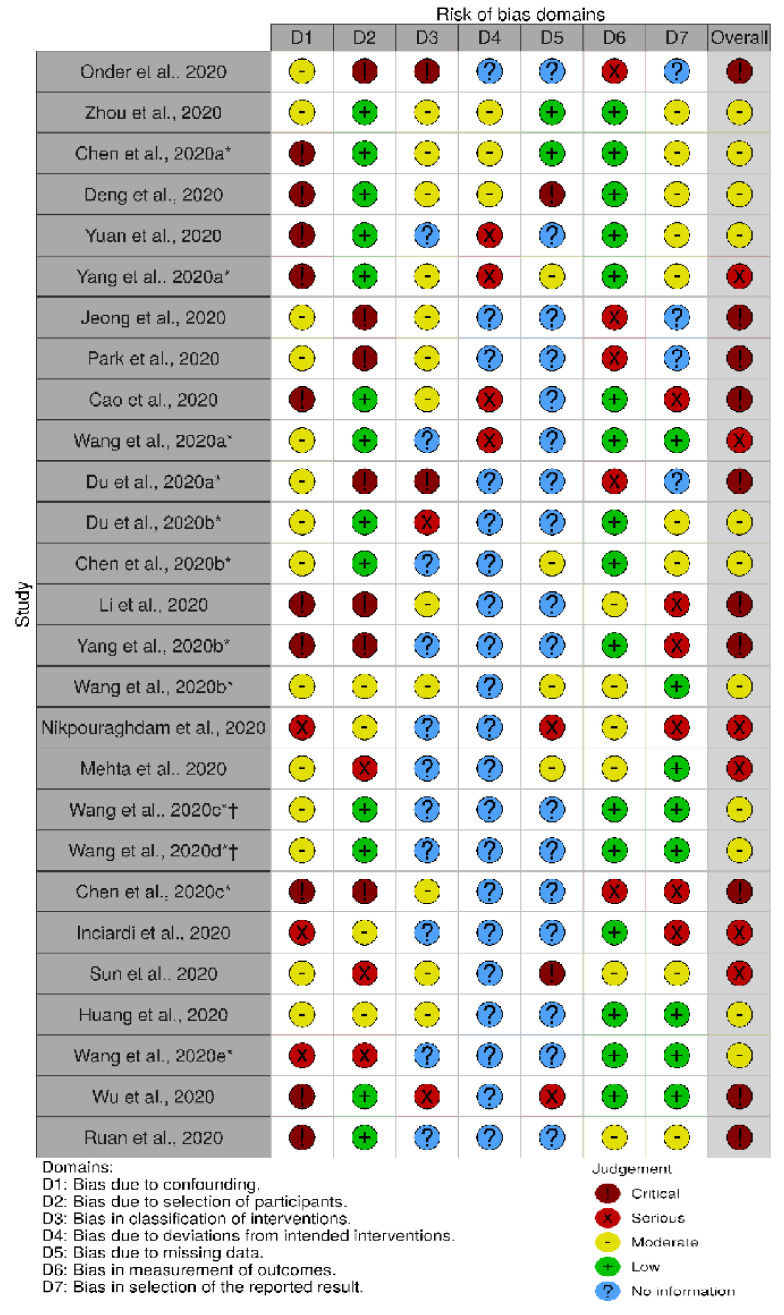
Risk of bias plot. * They are all different homonymous authors. † Two different cohorts from the same study.

**Figure 3 jcm-09-02685-f003:**
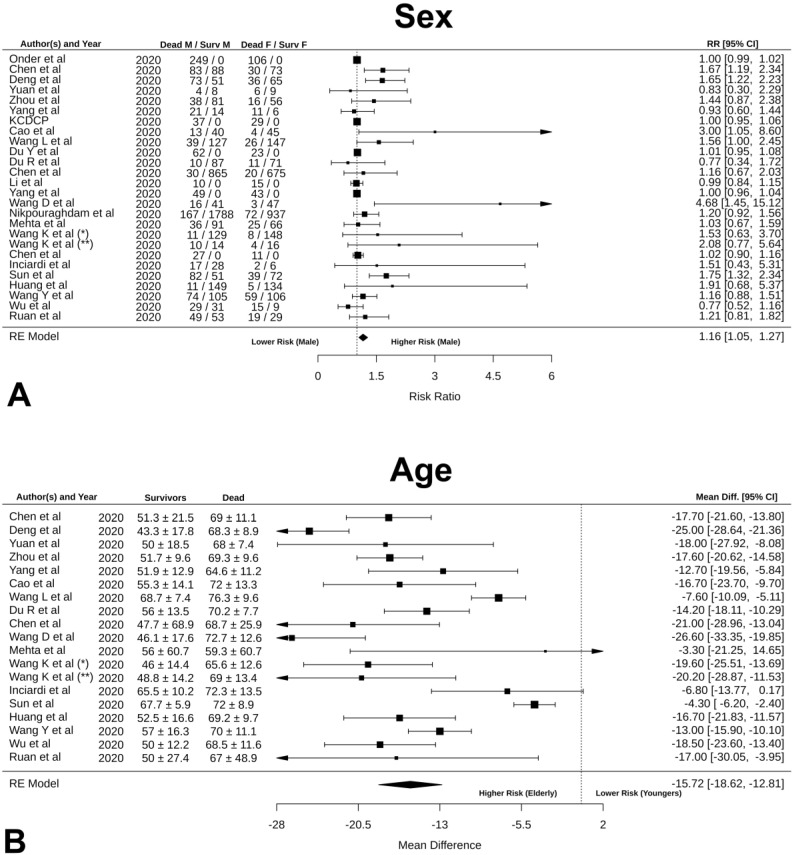
Forrest plot of (**A**) sex; (**B**) age.

**Figure 4 jcm-09-02685-f004:**
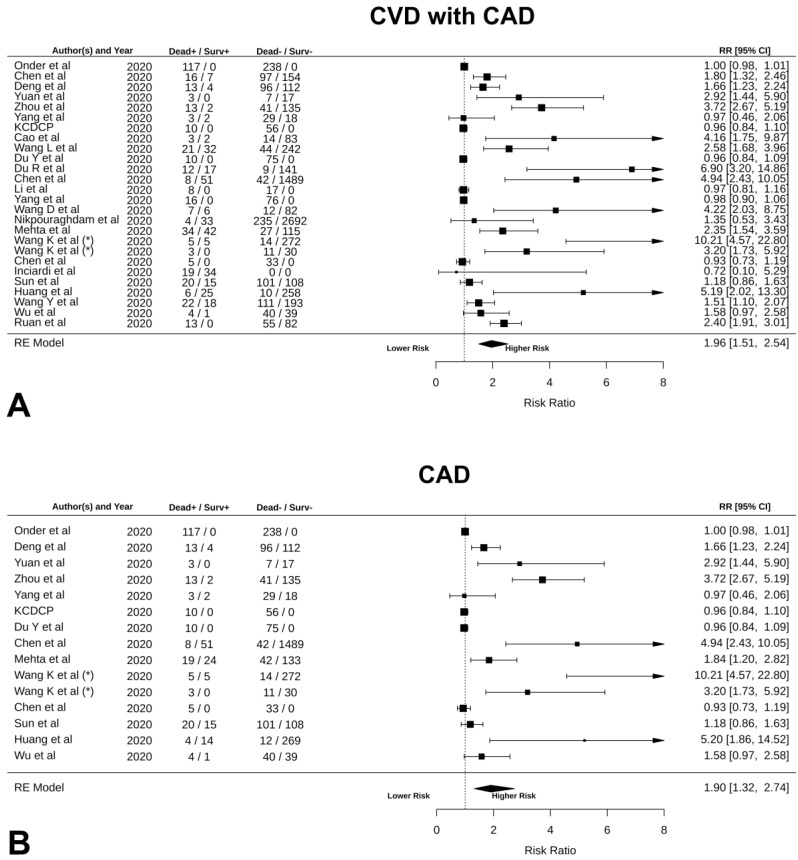
(**A**) CVD + CAD vs. patients without CVD; (**B**) patients with CAD vs. patients without CAD; (**C**) patients with CVD without CAD vs. patients without CVD; (**D**) patients with hypertension vs. patients without hypertension; (**E**) patients with cerebrovascular diseases vs. patients without cerebrovascular diseases; (**F**) patients with diabetes mellitus vs. patients without diabetes mellitus.

**Table 1 jcm-09-02685-t001:** Patients’ baseline characteristics ^1^.

Author, Year	Study Design	No. of Patients			F/M		Age		CVD		CAD		Hypertension		Diabetes		Cerebrovascular Disease	
		Overall	S	NS	S	NS	S	NS	S	NS	S	NS	S	NS	S	NS	S	NS
Onder et al., 2020 [21]	–	355	0 (0)	355 (100)	0	106/249	–	79.5 ± 8.1	0 (0)	117 (33)	0 (0)	117 (33)	–	–	0 (0)	126 (35.5)	0 (0)	34 (9.6)
Zhou et al., 2020 [29]	MRCS	191	137 (71.7)	54 (28.3)	56/81	16/38	52.0 (45.0–58.0)	69.0 (63.0–76.0)	2 (1.5)	13 (24.1)	2 (1.5)	13 (24.1)	32 (23.4)	26 (48.2)	19 (13.9)	17 (31.5)	–	–
Chen et al., 2020 * [11]	RCsSr	274	161 (58.8)	113 (41.2)	73/88	30/83	51.0 (37.0–66.0)	68.0 (62.0–77.0)	7 (4.3)	16 (14.2)	–	–	39 (24.2)	54 (47.8)	23 (14.3)	24 (21.3)	0 (0)	4 (3.5)
Deng et al., 2020 [13]	RCS	225	116 (51.6)	109 (48.4)	65/51	36/73	40.0 (33.0–57.0)	69.0 (62.0–74.0)	4 (3.4)	13 (11.9)	4 (3.4)	13 (11.9)	18 (15.5)	40 (36.7)	9 (7.8)	17 (15.6)	–	–
Yuan et al., 2020 [28]	RCS	27	17 (63.0)	10 (37.0)	9/8	6/4	55.0 (35.0–60.0)	68.0 (63.0–73.0)	0 (0)	3 (30.0)	0 (0)	3 (30.0)	0 (0)	5 (50.0)	0 (0)	6 (60.0)	0 (0)	1 (10.0)
Yang et al., 2020 * [27]	SROS	52	20 (38.5)	32 (61.5)	6/14	11/21	51.9 ± 12.9	64.6 ± 11.2	2 (10.0)	3 (9.4)	2 (10.0)	3 (9.4)	–	–	2 (10.0)	7 (21.9)	0 (0)	7 (21.9)
Jeong et al., 2020 (KCDCP) [33]		66	0 (0)	66 (100)	0	29/37	–	77 (35–93)	0 (0)	10 (15.2)	0 (0)	10 (15.2)	0 (0)	30 (45.5)	0 (0)	23 (34.8)	0 (0)	5 (7.6)
Park et al., 2020 (KSID) [34]		54	0 (0)	54 (100)	–	–	–	–	–	–	–	–	–	–	–	16 (29.6)	–	–
Cao et al., 2020 [9]	–	102	85 (83.3)	17 (16.7)	45/40	4/13	53.0 (47.0–66.0)	72.0 (63.0–81.0)	2 (2.4)	3 (17.6)	–	–	17 (20.0)	11 (64.7)	5 (5.9)	6 (35.3)	3 (3.5)	3 (17.6)
Wang et al., 2020 * [25]	RCS	339	274 (80.2)	65 (19.8)	147/127	26/39	68.0 (64.0–74.0)	76.0 (70.0–83.0)	32 (11.7)	21 (32.3)	–	–	106 (38.7)	32 (49.2)	43 (15.7)	11 (16.9)	11 (4.0)	10 (15.4)
Du et al., 2020 * [15]	ROS	85	0 (0)	85 (100)	0	23/62	–	65.8 ± 14.2	0 (0)	10 (11.8)	0 (0)	10 (11.8)	0 (0)	32 (37.6)	0 (0)	19 (22.4)	0 (0)	7 (8.2)
Du et al., 2020 * [14]	PCS	179	158 (88.3)	21 (11.7)	71/87	11/10	56.0 ± 13.5	70.2 ± 7.7	17 (10.8)	12 (57.1)	–	–	45 (28.5)	13 (61.9)	27 (17.1)	6 (28.6)	–	–
Chen et al., 2020 * [10]	RCS	1590	1540 (96.9)	50 (3.1)	675/865	20/30	48.0 (1.0–94.0) †	69.0 (51.0–86.0) †	51 (3.3)	8 (16.0)	51 (3.3)	8 (16.0)	241 (15.6)	28 (56.0)	117 (7.6)	13 (26.0)	24 (1.6)	6 (12.0)
Li et al., 2020 [18]	ROS	25	0 (0)	25 (100)	0	15/10	–	73.0 (55.0–100.0)	0 (0)	8 (32.0)	–	–	0 (0)	16 (64.0)	0 (0)	10 (40.0)	0 (0)	4 (16.0)
Yang et al., 2020 * [26]	ROS	92	0 (0)	92 (100)	0	43/49	–	69.8 ± 14.5 (30.0–97.0)	0 (0)	16 (17.4)	–	–	0 (0)	51 (55.4)	0 (0)	13 (14.1)	–	–
Wang et al., 2020 * [23]	RCsSr	107	88 (82.2)	19 (17.8)	47/41	3/16	44.5 (35.0–58.8)	73.0 (64.0–81.0)	6 (6.8)	7 (36.8)	–	–	16 (18.2)	10 (52.6)	6 (6.8)	5 (26.3)	3 (3.4)	3 (15.8)
Nikpouraghdam et al., 2020 [20]	SRCS	2964	2725 (91.9)	239 (8.1)	937/1788	72/167	–	65.0 (57.0–75.0)	33 (1.2)	4 (1.7)	–	–	51 (1.9)	8 (3.3)	102 (3.7)	11 (4.6)	–	–
Mehta et al., 2020 [19]	SROS	218	157 (72.0)	61 (28.0)	66/91	25/36	66.0 (10.0–92.0)	76.0 (10.0–92.0)	42 (26.8)	34 (55.7)	24 (15.3)	19 (31.1)	100 (63.7)	47 (77.0)	53 (33.8)	27 (44.3)	–	–
Wang et al., 2020a *‡ [24]	–	296	277 (93.6)	19 (6.4)	148/129	8/11	46.0 ± 14.4	65.6 ± 12.6	5 (1.8)	5 (26.3)	5 (1.8)	5 (26.3)	33 (11.9)	9 (47.4)	24 (8.7)	6 (31.6)	4 (1.4)	3 (15.8)
Wang et al., 2020b *‡ [24]	–	44	30 (68.2)	14 (31.8)	16/14	4/10	48.8 ± 14.2	69.0 ± 13.4	0 (0)	3 (21.4)	0 (0)	3 (21.4)	7 (23.3)	4 (28.6)	5 (16.7)	4 (28.6)	1 (3.3)	1 (7.1)
Chen et al., 2020 * [12]	RCS	38	0 (0)	38 (100)	0	11/27	–	70.0 (36.0-89.0)	0 (0)	5 (13.2)	0 (0)	5 (13.2)	0 (0)	15 (39.5)	0 0 (0)	11 (28.9)	0 (0)	4 (10.5)
Inciardi et al., 2020 [17]	–	53	34 (64.2)	19 (35.8)	6/28	2/17	65.5 ± 10.2	72.3 ± 13.5	34 (100)	19 (100)	–	–	27 (79.4)	13 (68.4)	7 (20.0)	9 (47.4)	–	–
Sun et al., 2020 [22]	RCC	244	123 (50.4)	121 (49.6)	72/51	39/82	67.0 (64.0–72.0)	72.0 (66.0–78.0)	15 (12.9)	20 (16.5)	15 (12.9)	20 (16.5)	62 (50.4)	76 (62.8)	24 (19.5)	27 (22.3)	–	–
Huang et al., 2020 [16]	RCS	299	283 (94.6)	16 (5.4)	134/149	5/11	52.5 ± 16.6	69.2 ± 9.7	25 (8.8)	6 (37.5)	14 (4.9)	4 (25.0)	63 (22.3)	11 (68.8)	31 (11.0)	4 (25.0)	–	–
Wang et al., 2020 * [30]	–	344	211 (61.3)	133 (38.7)	106/105	59/74	57.0 (47.0–69.0)	70.0 (62.0–77.0)	18 (8.5)	22 (16.5)	–	–	72 (34.1)	69 (51.9)	34 (16.1)	30 (22.6)	–	–
Wu et al., 2020 [31]	RCS	84	40 (47.6)	44 (52.4)	9/31	15/29	50.0 (40.3–56.8)	68.5 (59.3–75.0)	1 (2.5)	4 (9.1)	1 (2.5)	4 (9.1)	7 (17.5)	16 (36.4)	5 (10.0)	11 (25.0)	–	–
Ruan et al., 2020 [32]	MRCS	150	82 (54.7)	68 (45.3)	29/53	19/49	50.0 (44.0–81.0)	67.0 (15.0–81.0)	0 (0)	13 (19.1)	–	–	23 (28.0)	29 (42.6)	13 (15.9)	12 (17.6)	5 (6.1)	7 (10.3)

^1^ Values are expressed as mean ± standard deviation, median (interquartile range) or number (%). Abbreviations: CAD = coronary artery disease, CVD = cardio-vascular disease, F = females, M = males, MRCS = multi-center retrospective cohort study, NS = non-survivors, PCS = prospective cohort study, RCC = retrospective case control, RCS = retrospective cohort study, RCsS r= retrospective case series, S = survivors, SRCS = single-centered retrospective cohort study, SROS = single-centered retrospective observational study. * They are all different homonymous authors, † range, ‡ two different cohorts from the same study.

**Table 2 jcm-09-02685-t002:** Incidence rate ration (IRR), heterogeneity test and Egger’s test results ^1^.

	Mean Difference		Relative Risk		Heterogeneity		Publication Bias	
	MD [95% CI]	*p*-Value	RR [95% CI]	*p*-Value	I^2^ (%)	*p*-Value	Egger’s Intercept [95% CI]	*p*-Value
**Sex**	NA	NA	1.16 [1.05, 1.27]	0.003	90.79	<0.0001	2.67 [−0.01, 0.03]	0.002
**Age**	−15.72 [−18.62, 12.81]	<0.0001	NA	NA	86.74	<0.0001	−6.44 [−12.05, 0.81]	0.00
**CVD with CAD**	NA	NA	1.96 [1.51, 2.54]	<0.0001	97.87	<0.0001	−0.03 [−0.06, 0.01]	0.00
**CAD**	NA	NA	1.90 [1.32, 2.74]	0.0005	97.79	<0.0001	−0.02 [−0.06, 0.01]	0.00
**CVD without CAD**	NA	NA	2.03 [1.41, 2.92]	0.0002	93.25	<0.0001	−0.11 [−0.34, 0.13]	0.00
**Hypertension**	NA	NA	1.73 [1.37, 2.19]	<0.0001	98.01	<0.0001	−0.08 [−0.13, 0.03]	0.00
**Cerebrovascular diseases**	NA	NA	1.76 [1.25, 2.50]	<0.0001	98.01	<0.0001	−0.07 [−0.16, 0.01]	0.01
**Diabetes**	NA	NA	1.59 [1.25, 2.02]	<0.0001	98.71	<0.0001	−0.02 [−0.04, 0.00]	0.00

^1^ Abbreviations: CAD = coronary artery disease, CI = confidence interval, CVD = cardiovascular disease, MD = mean difference, NA = not applicable, RR = relative risk.

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
