# Peer review of "Quantification of Death Risk in Relation to Sex, Pre-Existing Cardiovascular Diseases and Risk Factors in COVID-19 Patients: Let’s Take Stock and See Where We Are"

_jcm, 2020, doi:10.3390/jcm9092685_

Round 1
Reviewer 1 Report
The manuscript entitled “Quantification of Death Risk in Relation to Sex, Pre-existing Cardiovascular Diseases and Risk Factors in COVID-19 Patients: Let's Take Stock and See Where We Are” by Amalia Ioanna Moula et al. demonstrates that sex, age, presence of coronary artery disease and/or other types of cardiovascular diseases, hypertension, cerebrovascular diseases and diabetes mellitus increase mortality in patients with COVID-19. This research is especially important in a time of the COVID-19 pandemic.
The authors’ claims are logically stated with the careful statistical analysis. Also, the limitation of the study is addressed by the authors in a cautious manner.
Below are minor comments:
Discussion about hydroxychloroquine and azithromycin is not directly related to the results of the authors’ study.
Line 281. “Therefore, also intravenous fluids administration should also receive attention by clinicians.”
Maybe the first “also” is not necessary?
Line 287. “This meta-analysis has some limitations that need to addressed”
Need to be addressed?
Reviewer 2 Report
This is an interesting paper. Addressing the following should improve it.
Minor comment:
1. The difference between CVD + coronary artery disease and CVD-CAD is at first glance unclear. Change CVD-CAD to CVD without CAD.
Major comments:
1. There is strong evidence that serious heart conditions, such as heart failure, coronary artery disease, or cardiomyopathies are linked with severe outcomes from Covid-19. Older age, male sex, coronary artery disease, cardiovascular disease and diabetes have been identified as risk factors for death. You conclude “Our results demonstrate that sex, age, presence of CAD and/or other types of CVD, hypertension, cerebrovascular diseases and diabetes mellitus increase mortality in patients with COVID-19”. Are you findings really new?
2. There is mixed evidence on cerebrovascular disease. You seem uncertain about whether your data provides clarity. Your statistics suggest it is a greater risk than diabetes. Is there a reason why you state that “that cerebrovascular disease might be a strong predictor of death”?
3. Does your data shed any light on type 1 versus type 2 diabetes mellitus as risk factors for death?
4. The section, Prognostic factors for mortality (lines 167-193), is very statistically dense and hard to read. Try simplifying it without losing its impact.
5. Figures 2 and 3 are very small and essentially inscrutable because of their size.
6. The explanations of potential pathological mechanisms are superb. However, the discussion might be more readable if it were divided into subsections by individual risk factors and the proposed mechanisms making them high risk.
7. The discussion of the ineffective agents hydroxychloroquine and azithromycin is irrelevant and should be deleted.
Round 2
Reviewer 2 Report
The manuscript is improved. The following comments are primarily aimed at improving the manuscript’s grammar and syntax.
Minor comments:
1. On lines 51-53 did you mean to write: Nevertheless, the exact mechanisms by which COVID-19 affects the cardiovascular system and mortality are not yet well understood, despite evidence that such a connection is accumulating [4,5]?
2. On line 72, did you intend to duplicate "covid-19" OR "covid 19"?
3. On line 85, change: “on survived and non-survived patients” to “on surviving and non-surviving patients”.
4. On line 123, change: non-survived to non-survivors.
5. On line 127, change: form to from.
6. On line 137, change: “patient’s cohort comprises of” to “patient cohort included”.
7. On line 211, identify IRR as incidence rate ratio.
8. On line 221 change: “In particular, presence of CAD is able to increase the risk of mortality …” to “In particular, the presence of CAD increased the risk of mortality…”.
9. On line 224, delete the word “also”.
10. On lines 228-229 change: “We found that non-survivors were more likely affected by hypertension, being them 1.73-fold more at risk of mortality than patients without hypertension” to “We found that non-survivors were more likely affected by hypertension, having a 1.73-fold greater risk of mortality than patients without hypertension”.
11. On line 233 delete “given the strong evidence of this study”.
12. On lines 268-272 change: “It has been widely reported an increased risk of mortality for male COVID-19 patients with the number of men is 2.4 times that of women in the deceased patients. This disproportionate death ratio in men was explained by higher incidence of pre-existing disease, higher risk behaviors, occupational exposure, behavioral and social differences that favor women and high levels of androgens in men [35]” to “An increased risk of mortality for male COVID-19 patients (2.4 times that of women) has been widely reported. This disproportionate death ratio in men was explained by a higher incidence of pre-existing disease, higher risk behaviors, occupational exposure, high levels of androgens in men, and behavioral/social differences that favor women and [35]”.
13. On line 278, change: “that had” to “with”.
14. On line 280, change: “it” to “there”.
15. On line 283, change: “Primary” to “Primarily”.
16. On line 291, change: “cells” to “cell”.
17. On lines 316-318, change: “ADAM17 causes the cleavage of the
18. ACE2 receptors making them not responsive anymore to the negative feedback exerted by the activation of the renin–angiotensin–aldosterone system” to “ADAM17 causes cleavage of the ACE2 receptors making them unresponsive to the negative feedback exerted by the activation of the renin–angiotensin–aldosterone system”.
19. On line 320, delete “the”.
20. On line 321, change: “increasing the ratio oxygen demand/supply” to “increasing the oxygen demand/supply ratio”.
21. On line 323, change: “rose” to “raised”.
22. On lines 329-335, change: ”…the true risk increment brought by hypertension in COVID-19 patients with CVD. However, it has been proposed that patients with heart failure and hypertension receive ACE inhibitors and type-I receptor blockers (ARBs), which contribute to the upregulation of ACE2 receptors. This increases susceptibility to contract COVID-19 and the mechanisms mentioned above of vulnerability towards mortality take place. For this reason, some authors have suggested the use of alternative antihypertensive medication during the pandemics, such as calcium channel blockers [49]” to the true incremental risk associated with hypertension in COVID-19 patients with CVD. However, it has been proposed that when patients with heart failure and hypertension receive ACE inhibitors and type-I receptor blockers (ARBs), these agents contribute to the upregulation of ACE2 receptors. This increases susceptibility to contracting COVID-19. The mechanisms mentioned help explain their vulnerability to mortality. For this reason, some authors have suggested the use of alternative antihypertensive medication during the pandemic, such as calcium channel blockers [49]”.
Major comments:
1. Figure 4 remains small and difficult to see.
2. Under Study Limitations, the sentence “This can easily happen when analyzing a consistent number of studies on events such as sepsis and traumas, which encompass a more different population” is inscrutable. It can be deleted without impacting the section.
3. Under Study Limitations: Since covid 19 has only been around since 12/31/19, the concept of RR increase per year is confusing. Do you mean, for instance, the relative risk of dying at age 58 versus age 65? Please clarify.
Author Response
Changes and replies to the comments of the reviewers for the manuscript entitled: “Quantification of Death Risk in Relation to Sex, Pre-existing Cardiovascular Diseases and Risk Factors in COVID-19 Patients: Let's Take Stock and See Where We Are” by Amalia Ioanna Moula et al.
Comment:
The manuscript is improved. The following comments are primarily aimed at improving the manuscript’s grammar and syntax.
Reply:
We would like to thank the reviewer for the detailed and in-depth comments that help improve this manuscript.
Minor comments:
Comment:
- On lines 51-53 did you mean to write: Nevertheless, the exact mechanisms by which COVID-19 affects the cardiovascular system and mortality are not yet well understood, despite evidence that such a connection is accumulating [4,5]?
Reply:
We agree with the comment and rephrased the sentence.
Changes:
We rephrased as: Nevertheless, the exact mechanisms by which COVID-19 affects the cardiovascular system and mortality are not yet well understood, despite indications on the nature of such a connection are accumulating despite accumulating evidence that such a connection exists [4,5].
Comment:
- On line 72, did you intend to duplicate "covid-19" OR "covid 19"?
Reply:
The correct term covid-19 was included in the search together with the spelling “covid 19”. We tried to include in our search any possible spellings of COVID-19. We actually meant “covid 19” and this term was used additionally to the correct COVID-19 in order to avoid missing any relevant papers.
Changes: No change.
Comment:
- On line 85, change: “on survived and non-survived patients” to “on surviving and non-surviving patients”.
Reply: We agree.
Change: We changed it accordingly to “on surviving and non-surviving patients”
Comment:
- On line 123, change: non-survived to non-survivors.
We changed it accordingly.
Reply: We agree.
Change: We changed it accordingly to “non-survivors”
Comment:
- On line 127, change: form to from.
Reply: We agree.
Change: We changed it accordingly to “from”
Comment:
- On line 137, change: “patient’s cohort comprises of” to “patient cohort included”.
Reply: We agree.
Change: We changed it accordingly to “patient cohort included”
Comment:
- On line 211, identify IRR as incidence rate ratio.
Reply: We agree.
Change: We changed it to “Incidence rate ratio (IRR)”
Comment:
- On line 221 change: “In particular, presence of CAD is able to increase the risk of mortality …” to “In particular, the presence of CAD increased the risk of mortality…”.
Reply: We agree.
Change: We changed it accordingly to “In particular, the presence of CAD increased the risk of mortality”
Comment:
- On line 224, delete the word “also”.
Reply: We agree.
Change: The word “also” was deleted.
Comment:
- On lines 228-229 change: “We found that non-survivors were more likely affected by hypertension, being them 1.73-fold more at risk of mortality than patients without hypertension” to “We found that non-survivors were more likely affected by hypertension, having a 1.73-fold greater risk of mortality than patients without hypertension”.
Reply: We agree.
Change:The phrase was changed accordingly to “We found that non-survivors were more likely affected by hypertension, having a 1.73-fold greater risk of mortality than patients without hypertension”..
Comment:
- On line 233 delete “given the strong evidence of this study”.
Reply: We agree.
Change: The phrase was deleted.
Comment:
- On lines 268-272 change: “It has been widely reported an increased risk of mortality for male COVID-19 patients with the number of men is 2.4 times that of women in the deceased patients. This disproportionate death ratio in men was explained by higher incidence of pre-existing disease, higher risk behaviors, occupational exposure, behavioral and social differences that favor women and high levels of androgens in men [35]” to “An increased risk of mortality for male COVID-19 patients (2.4 times that of women) has been widely reported. This disproportionate death ratio in men was explained by a higher incidence of pre-existing disease, higher risk behaviors, occupational exposure, high levels of androgens in men, and behavioral/social differences that favor women and [35]”.
Reply: We agree.
Change: The phrase was changed as recommended.
Comment:
- On line 278, change: “that had” to “with”.
Reply: We agree.
Change: The change was made as recommended.
Comment:
- On line 280, change: “it” to “there”.
Reply: We think that the reviewer meant line 272 of the first revision of the manuscript “where it has been a turn in sex-specific differences” and we changed it to “whether there has been a turn in sex-specific differences”. Changing the word “it” in line 280 would make the sentence less understandable.
Change: We changed “it” in the phrase “where it has been” to “whether there has ”.
Comment:
- On line 283, change: “Primary” to “Primarily”.
Reply: We agree.
Change: The change was made as recommended to “Primarily”.
Comment:
- On line 291, change: “cells” to “cell”.
Reply: We agree.
Change: The change was made as recommended to “cell”.
Comment:
- On lines 316-318, change: “ADAM17 causes the cleavage of the ACE2 receptors making them not responsive anymore to the negative feedback exerted by the activation of the renin–angiotensin–aldosterone system” to “ADAM17 causes cleavage of the ACE2 receptors making them unresponsive to the negative feedback exerted by the activation of the renin–angiotensin–aldosterone system”.
Reply: We agree.
Change: The change was made as recommended to “ADAM17 causes cleavage of the ACE2 receptors making them unresponsive to the negative feedback exerted by the activation of the renin–angiotensin–aldosterone system”.
Comment:
- ACE2 receptors making them not responsive anymore to the negative feedback exerted by the activation of the renin–angiotensin–aldosterone system” to “ADAM17 causes cleavage of the ACE2 receptors making them unresponsive to the negative feedback exerted by the activation of the renin–angiotensin–aldosterone system”.
Reply: We agree.
Change: We changed the phrase to “ADAM17 causes cleavage of the ACE2 receptors making them unresponsive to the negative feedback exerted by the activation of the renin–angiotensin–aldosterone system.”
Comment:
- On line 320, delete “the”.
Reply: We agree.
Change: The word “the” was deleted.
Comment:
- On line 321, change: “increasing the ratio oxygen demand/supply” to “increasing the oxygen demand/supply ratio”.
Reply: We agree.
Change: The change was made to “increasing the oxygen demand/supply ratio”
Comment:
- On line 323, change: “rose” to “raised”.
Reply: We agree.
Change: The change was made to “raised”
Comment:
- On lines 329-335, change: ”…the true risk increment brought by hypertension in COVID-19 patients with CVD. However, it has been proposed that patients with heart failure and hypertension receive ACE inhibitors and type-I receptor blockers (ARBs), which contribute to the upregulation of ACE2 receptors. This increases susceptibility to contract COVID-19 and the mechanisms mentioned above of vulnerability towards mortality take place. For this reason, some authors have suggested the use of alternative antihypertensive medication during the pandemics, such as calcium channel blockers [49]” to the true incremental risk associated with hypertension in COVID-19 patients with CVD. However, it has been proposed that when patients with heart failure and hypertension receive ACE inhibitors and type-I receptor blockers (ARBs), these agents contribute to the upregulation of ACE2 receptors. This increases susceptibility to contracting COVID-19. The mechanisms mentioned help explain their vulnerability to mortality. For this reason, some authors have suggested the use of alternative antihypertensive medication during the pandemic, such as calcium channel blockers [49]”.
Reply: We agree.
Change: The change was made to “the true incremental risk associated with hypertension in COVID-19 patients with CVD. However, it has been proposed that when patients with heart failure and hypertension receive ACE inhibitors and type-I receptor blockers (ARBs), these agents contribute to the upregulation of ACE2 receptors. This increases susceptibility to contracting COVID-19. The mechanisms mentioned help explain their vulnerability to mortality. For this reason, some authors have suggested the use of alternative antihypertensive medication during the pandemic, such as calcium channel blockers [49]”.
Major comments:
Comment:
- Figure 4 remains small and difficult to see.
Reply: The figure was divided in three sub-figures, namely 4a. 4b. and 4c. in order to be easier to read. Subsequently the legend was also divided.
Change:
Figure 4. a. A. CVD+CAD vs patients without CVD. B. Patients with CAD vs patients without CAD.
Figure 4. b. C. Patients with CVD without CAD vs patients without CVD. D. Patients with hypertension vs patients without hypertension.
Figure 4. c. E. Patients with cerebrovascular diseases vs patients without cerebrovascular diseases. F. Patients with diabetes mellitus vs patients without diabetes mellitus.
Comment:
- Under Study Limitations, the sentence “This can easily happen when analyzing a consistent number of studies on events such as sepsis and traumas, which encompass a more different population” is inscrutable. It can be deleted without impacting the section.
Reply: We agree.
Change: The phrase was deleted.
Comment:
- Under Study Limitations: Since covid 19 has only been around since 12/31/19, the concept of RR increase per year is confusing. Do you mean, for instance, the relative risk of dying at age 58 versus age 65? Please clarify.
Reply: We made changes in order to clarify.
Change:
We changed the “per year” to “with age”. Also we changed the word years in the phrase “the difference in years” with the word age to “the difference in age”. The paragraph was changed to : “Finally, in the reports there was reported age cut-off, therefore it was not possible to examine the RR increase per year with age. The only data attainable was the difference in years age between survivors and not survivors”
